# Assessment of the Analytic Burned Area Index for Forest Fire Severity Detection Using Sentinel and Landsat Data

Rentao Guo [1], Jilin Yan [1], He Zheng [1,2] and Bo Wu [1,3,*]

1 School of Geography and Environment, Jiangxi Normal University, Nanchang 330022, China; guorentao@jxnu.edu.cn (R.G.); 202350000008@jxnu.edu.cn (J.Y.); zheng_he@jxnu.edu.cn (H.Z.)
2 School of Geomatics and Geoinformation, Jiangxi College of Applied Technology, Ganzhou 341000, China
3 Key Laboratory of PoYang Lake Wetland and Watershed Research, Ministry of Education, Jiangxi Normal University, Nanchang 330022, China
* Correspondence: wubo@jxnu.edu.cn

**Abstract:** The quantitative assessment of forest fire severity is significant for understanding the changes in ecological processes caused by fire disturbances. As a novel spectral index derived from the multi-objective optimization algorithm, the Analytic Burned Area Index (ABAI) was originally designed for mapping burned areas. However, the performance of the ABAI in detecting forest fire severity has not been addressed. To fill this gap, this study utilizes a ground-based dataset of fire severity (the composite burn index, CBI) to validate the effectiveness of the ABAI in detecting fire severity. First, the effectiveness of the ABAI regarding forest fire severity was validated using uni-temporal images from Sentinel-2 and Landsat 8 OLI. Second, fire severity accuracy derived from the ABAI with bi-temporal images from both sensors was evaluated. Finally, the performance of the ABAI was tested with different sensors and compared with representative spectral indices. The results show that (1) the ABAI demonstrates significant advantages in terms of accuracy and stability in assessing fire severity, particularly in areas with large numbers of terrain shadows and severe burn regions; (2) the ABAI also shows great advantages in assessing regional forest fire severity when using only uni-temporal remotely sensed data, and it performed almost as well as the dNBR in bi-temporal images. (3) The ABAI outperforms commonly used indices with both Sentinel-2 and Landsat 8 data, indicating that the ABAI is normally more generalizable and powerful and provides an optional spectral index for fire severity evaluation.

**Keywords:** forest fires; spectral indices; ABAI; fire severity; Sentinel-2; Landsat 8 OLI

## 1. Introduction

Forest fires have a significant impact on forest ecosystems, altering their biodiversity and ecological sustainability, leading to an ecological imbalance [1–3]. The term "fire severity" is generally defined as the extent to which a location is altered or damaged by a fire in the "Glossary of Wildland Fire Terminology" [4]. The quantitative assessment of fire severity is an important aspect of studying the response mechanisms of terrestrial ecosystems to fire and is important for understanding the ecological processes caused by forest fires [5,6]. However, the severity of fire events often varies significantly from region to region, and a lack of consensus on methods for assessing fire severity limits our understanding of the role of forest fires in various terrestrial ecosystems. Therefore, it is urgently necessary to improve existing methods or construct new spectral indices for the assessment of fire severity [7].

Satellite images provide spatially explicit information for the investigation of fire severity, and they have long been regarded as a valuable data source for monitoring biomass burning from the local to global scales, along with its dynamic characteristics [8,9]. Over the past half-century, wildfire researchers have developed various qualitative and quantitative methods for fire severity based on field surveys and the remote sensing monitoring of

burn areas [10]. Currently, one of the most commonly used methods of assessing fire severity is the comprehensive use of satellite remote sensing spectral indices in combination with the Composite Burn Index (CBI), which was designed as an operational field-based methodology for burn severity assessment that can rate the average burn severity within sample areas [11,12]. The CBI has been frequently employed in many studies to validate different remote sensing products, as well as for the comprehensive assessment of fire severity in various forest ecosystems [13–15].

Over the past two decades, numerous spectral indices have been developed to quantify fire severity, and each index presents its characteristics in local studies. These indices differ in the combination of spectral bands used, resulting in performance differences for assessing fire severity. For instance, the Normalized Burn Ratio (NBR), which utilizes NIR (near infrared) and SWIR (shortwave infrared) wavelengths to delineate burned areas, is one of the most commonly used indices for the measurement of fire severity [16]. Among the most commonly used optical features, vegetation indices play a major role in studies of the remote sensing of forest burn areas and burn severity. The Normalized Difference Vegetation Index (NDVI), calculated by combining the red and NIR bands, correlates well with fire severity. It is the most widely used vegetation index for assessing fire severity [17,18]. The Soil-Adjusted Vegetation Index (SAVI) uses NIR and red wavelengths to mitigate spectral interference from soil and has also been widely used for assessing fire severity [19,20]. Similarly, the burned area index (BAI) employs the NIR and red wavelengths to separate burned areas by identifying the charcoal signal in post-fire images, which integrates the minimum reflectance of burnt vegetation in the NIR and the maximum in the red bands [21]. The Char Soil Index (CSI) is calculated using the ratio of the NIR and LSWIR spectral bands and is designed to detect black carbon signals to assess fire severity [22]. In addition, The Mid-Infrared Bi-Spectral Index (MIRBI) has high potential in separating burned areas from unburned areas, particularly at water–land boundaries, and is, therefore, also used in fire severity assessment research [23]. However, despite the availability of a variety of indices for assessing fire severity, there is currently no universally applicable spectral index to quantify the severity of fire based on satellite images. Moreover, the suitability of various spectral indices for assessing fire severity in different regions and whether the evaluation results meet accuracy requirements remain unclear. It is, therefore, essential to further investigate the effectiveness of different spectral indices for a comprehensive assessment of fire severity.

The Analytic Burned Area Index (ABAI) is a novel burned area index recently developed by our research team through a multi-objective optimization method. The construction of this index begins with an analysis of the spectral characteristics of existing burned area indices and the spectral differences between burned areas and other land cover types. For each land cover type, we formulate an objective function through linear combinations of band-to-band ratios. Subsequently, all objective functions and possible constraint equations are transformed into a multi-objective optimization problem, which is then solved using linear programming techniques. Finally, the optimized coefficients derived from the multi-objective problem yield the ABAI. Current research indicates that the ABAI has shown significant advantages in mapping burned areas based on Sentinel-2 imagery, particularly regarding confusing areas with water bodies and shadow cover [24]. However, its potential for assessing fire severity and its performance with other satellite sensors has yet to be studied by researchers.

In this context, we extend the role of the ABAI from mapping burned areas to detecting fire severity and aim to evaluate and compare the performance of the ABAI with other commonly used spectral indices for assessing forest fire severity using multiple satellite sensors. The obtained assessment results provide a scientific foundation for future studies on fire severity through remotely sensed technology. Moreover, this paper can contribute to the conservation of forest resources and the promotion of post-fire vegetation recovery.

## 2. Materials and Methods

### 2.1. Study Area

To comprehensively assess the applicability of the ABAI across different satellite sensors and regions, this study utilized remote sensing data from two types of satellite sensors in two different fire-prone study areas located at high and low latitudes. The first study area is situated in Ganzhou City, Jiangxi Province, China, with coordinates ranging from approximately 24°29′ to 27°09′ N latitude and 113°54′ to 116°38′ E longitude. This region falls within the low-latitude zone and a typical subtropical monsoon climate. It boasts a forest cover of 76%, with dominant tree species being Masson pine and Chinese fir. The topography is characterized by mountains, hills, and fault basins that traverse the city, making it predominantly mountainous and hilly [25,26]. The study area has a complex surface environment with frequent forest fires. Moreover, the presence of mountain shadows adds to the complexity of burned area detection. Therefore, this area was selected for our study. Within this study area, there were three specific fire events, including the Yinkeng Farm fire, which occurred in Yudu County, Ganzhou City, on 4 January 2021; the Changmianling fire, which started on 14 January 2021 in Ganxian County, Ganzhou City; and the Gooseback fire, which broke out on 15 January 2021 in Nankang District, Ganzhou City.

The second study area is located in Okanogan County, Washington, USA. Its coordinates range from approximately 51°00′ to 52°00′ N latitude and 120°16′ to 123°00′ W longitude. This region experiences a temperate oceanic climate and features gently rolling terrain, primarily consisting of broad mountain ranges and valleys. The state of Washington has a forest coverage rate of 53%, with dominant tree species including spruce, pine, and fir, along with a rich diversity of vascular plants [27,28]. To validate the applicability of the ABAI in high-latitude areas, this region was selected as the study area. The wildfire event within this study area occurred on 14 July 2014 in central Washington State, covering an area of over 2800 hectares. Figure 1 illustrates the geographic locations where the two study areas are located.

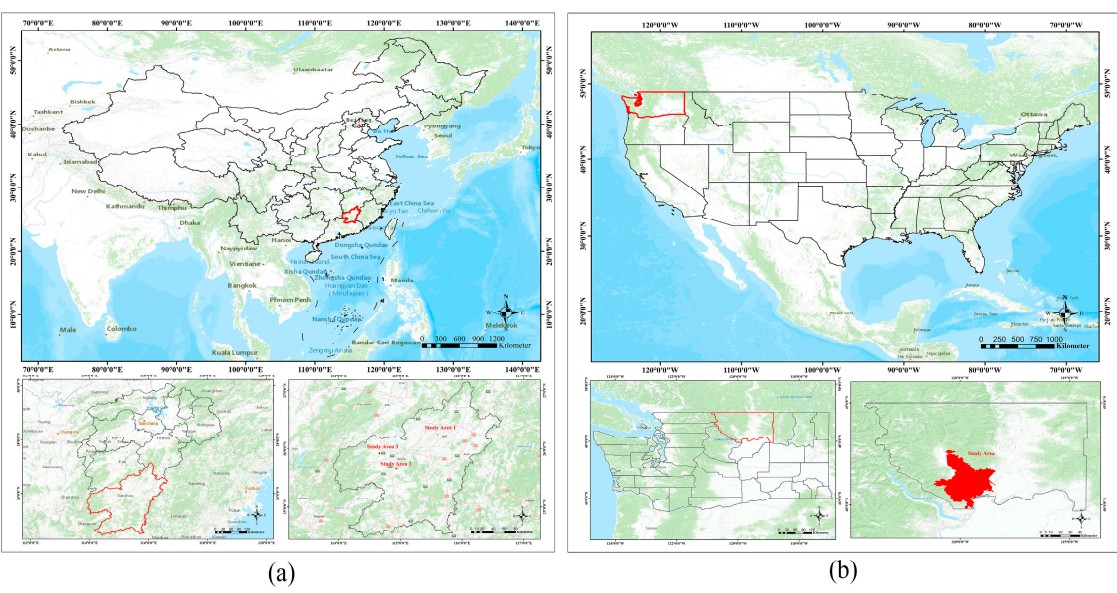

        (a)                       (b)

**Figure 1.** Locations of the two study areas. (**a**) Ganzhou City, Jiangxi Province, China. (**b**) Okanogan County, Washington, USA.

### 2.2. CBI-Based ABAI Computation and Assessment

To test the effectiveness of the ABAI in assessing fire severity, evaluations were conducted for both study areas using uni-temporal imagery data. Additionally, to assess the sensitivity of the ABAI in evaluating fire severity, assessments were performed for both

study areas using bi-temporal imagery data. Furthermore, to examine the universality of the ABAI in assessing fire severity across different satellite sensors, experiments were conducted using imagery data from two satellite sources: Sentinel-2 and Landsat 8.

The pre-processing of pre-fire and post-fire imagery data from Sentinel-2 and Landsat 8 OLI satellites was carried out using the SNAP software version 9.0.0 and the "SEN2COR" plugin tools to facilitate the computation of spectral indices. In this study, 70% of the ground-based composite burn index (CBI) data were randomly sampled for training samples. Threshold regression models, including linear, quadratic, and cubic polynomial models, were established to determine the relationship between the CBI and various spectral indices. The accuracy of these models was assessed based on the coefficient of determination ($R^2$) and the root mean square error (RMSE). The best-performing model was selected for the construction of fire severity thresholds. Subsequently, the remaining 30% of ground-based CBI sampling data were used to generate a confusion matrix, including overall accuracy and a kappa coefficient. To assess the accuracy and stability of the ABAI in fire severity classification, experiments were repeated five times by randomly shuffling the data between the training and validation sets. Figure 2 shows the technical framework of this paper.

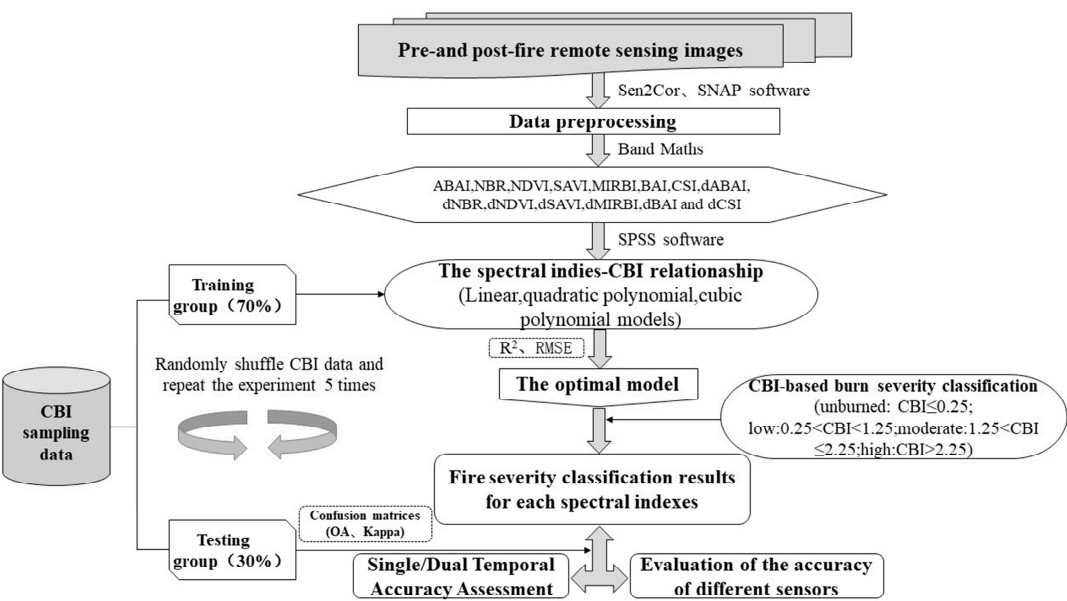

**Figure 2.** Flowchart of the methodology used in this study.

*2.3. Data Collection and Processing*

2.3.1. Remote Sensing Data

The remote sensing data used in this study originated from two types of sensors: Sentinel-2 and Landsat 8 OLI. Each sensor type provided pre- and post-fire remote sensing images of the study area. The Sentinel-2 satellite, provided by the European Space Agency (ESA), was launched on 23 June 2015. It carries a Multispectral Imager (MSI) capable of covering 13 spectral bands with spatial resolutions ranging from 10 to 60 m. These bands include the visible, near-infrared (VNIR), and short-wave infrared (SWIR) spectral ranges. For all the pre-and post-fire images, pre-processing was conducted using the "SEN2COR" plugin tool that ESA provided, including radiometric calibration and atmospheric correction. Subsequently, the spatial resolution of all bands was resampled to 10 m using bilinear interpolation in the SNAP software to facilitate further research.

The Landsat 8 satellite, operated by the United States Geological Survey (USGS), was launched on 11 February 2013. It carries the Operational Land Imager (OLI) and the Thermal Infrared Sensor (TIRS) and covers 9 spectral bands with a spatial resolution of 30 m. These bands include those in the VNIR and SWIR spectral ranges. The Landsat 8 OLI data used in this study were processed to the L1T (Level 1T) processing level, which includes

geometric and terrain correction. As a result, the image data only required radiometric calibration and atmospheric correction processing before further analysis.

The post-fire images for of the two study areas were selected as the closest images after the CBI sampling time, while the pre-fire images for the city of Ganzhou, China, were selected as the closest images to the time of the fire. The CBI data for Okanogan County, USA, was sampled one year after the fire; so, to ensure that the study was for the same season, imagery from one year before the sampling time was selected as pre-fire imagery. The extent of the study area for all selected images was not covered by cloud cover. Table 1 shows all the remote sensing image information used in the study.

**Table 1.** Data parameters and sources.

| Study Area | Sensor | Path/Row | Cloud Cover | Acquisition Date |
|---|---|---|---|---|
| Ganzhou | Sentinel-2A | N0209, R132 | 0% | 1 January 2021 |
| | Sentinel-2A | N0209, R132 | 62.6% | 21 January 2021 |
| | Sentinel-2B | N0209, R032 | 0% | 9 January 2021 |
| | Sentinel-2B | N0209, R032 | 0.1% | 19 January 2021 |
| | Landsat 8 OLI | 121, 042 | 40.7% | 27 December 2020 |
| | Landsat 8 OLI | 122, 042 | 0.3% | 18 February 2020 |
| | Landsat 8 OLI | 121, 042 | 0% | 12 January 2021 |
| | Landsat 8 OLI | 122, 042 | 0% | 19 January 2021 |
| Okanogan | Landsat 8 OLI | 045, 026 | 2.7% | 15 July 2014 |
| | Landsat 8 OLI | 045, 027 | 1.6% | 15 July 2014 |
| | Landsat 8 OLI | 045, 026 | 1.8% | 2 July 2015 |
| | Landsat 8 OLI | 045, 027 | 0.4% | 2 July 2015 |

2.3.2. CBI Data

The three fire events in Ganzhou City, Jiangxi Province, China, all occurred in January 2021. Field surveys of the burned areas were carried out within one month after the fire event by using the CBI field sampling method. During the field survey, sample plots measuring 30 m × 30 m were used. These plots were distributed in areas that exhibited varying topographic conditions and burn severity. Each plot was separated into four layers based on their vertical heights: layer A symbolizes surface combustible material and soil layers; layer B includes herbs, dwarf shrubs, and small trees (<1 m tall); layer C involves tall shrubs and trees (>1.5 m tall); and layer D comprises the forest canopy (>5 m) [29–31]. Each layer had 4–5 variables, with visual estimates ranging from 0 to 3 and fire severity ranging from low to high, classified as no burn, low burn severity, moderate burn severity, or high burn severity. The CBI values for the entire plot were calculated by combining the estimates of each layer.

The CBI formula for each layer is as follows:

$$CBI_i = \sum_{j=1}^{n} X_{ij}/n \tag{1}$$

where $CBI_i$ is the composite burn index in stratum i; $X_{ij}$ is the severity-rating score of factor j in stratum i; and n is the total number of factors in stratum i.

The CBI for each sample plot was calculated as follows:

$$CBI = \sum_{i=1}^{K} CBI_i/k \tag{2}$$

where CBI is the composite burn index of the investigated plots, and k is the total number of burned layers in the measured plots.

In January 2021, a total of 240 CBI data samples were collected in Ganzhou City, Jiangxi Province, China. These samples were randomly divided into two groups: 70% of the CBI data samples were used to construct the threshold regression model, while the remaining 30% were used to validate the accuracy of the classification results for various spectral indices (Table 2). Fire severity was classified into four levels: unburned (CBI ≤ 0.25), low

burn severity (0.25 < CBI $\leq$ 1.25), moderate burn severity (1.25 < CBI $\leq$ 2.25), and high burn severity (CBI > 2.25) [32,33].

**Table 2.** Statistics of the composite burn index (CBI) of Ganzhou City, Jiangxi Province, China.

| Severity Grade | Class Boundary, CBI | Number (Train/Test) | Minimum | Maximum | Mean | Standard Deviation |
|---|---|---|---|---|---|---|
| Unburned | [0, 0.25] | 40 (28/12) | 0 | 0.24 | 0.01 | 0.04 |
| Low | [0.25, 1.25] | 20 (14/6) | 0.3 | 1.25 | 1.03 | 0.17 |
| Moderate | [1.25, 2.25] | 90 (63/27) | 1.26 | 2.25 | 1.82 | 0.29 |
| High | [2.25, 3] | 90 (63/27) | 2.27 | 3 | 2.59 | 0.20 |

The wildfire in Okanogan County, Washington, occurred in July 2014. CBI sampling data and burned area data for this region were sourced from the United States Geological Survey (USGS), and CBI sampling occurred from May through June 2015. A total of 257 CBI sampling data points were provided, and similar to the previous study area, these were divided into two groups: 70% of the CBI data samples were used to build the threshold regression model, and the remaining 30% were used for validating the accuracy of the classification results for various spectral indices (Table 3). More detailed information on the CBI data can be found on the website https://www.sciencebase.gov/catalog/item/5d963705e4b0c4f70d110ee6, accessed on 10 June 2023. Figure 3 illustrates the location of CBI sampling sites in each study area.

**Table 3.** Statistics of the composite burn index (CBI) of Okanogan County, Washington, USA.

| Severity Grade | Class Boundary, CBI | Number (Train/Test) | Minimum | Maximum | Mean | Standard Deviation |
|---|---|---|---|---|---|---|
| Unburned | [0, 0.25] | 45 (32/13) | 0 | 0.24 | 0.02 | 0.06 |
| Low | [0.25, 1.25] | 99 (69/30) | 0.27 | 1.23 | 0.79 | 0.25 |
| Moderate | [1.25, 2.25] | 54 (38/16) | 1.3 | 2.21 | 1.72 | 0.30 |
| High | [2.25, 3] | 59 (41/18) | 2.3 | 2.9 | 2.60 | 0.20 |

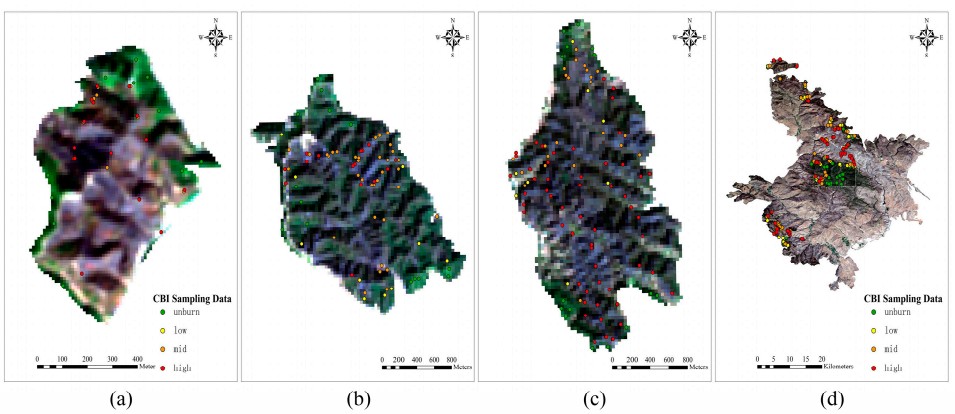

(a)     (b)     (c)     (d)

**Figure 3.** Location of CBI sampling sites in each study area. (**a**) Yinkeng Farm fire in Yudu County, Ganzhou City. (**b**) Qipanqiu Changmianling fire in Ganxian County, Ganzhou City. (**c**) Gooseback fire in Nankang District, Ganzhou City. (**d**) Okanogan County, Washington, USA.

*2.4. Spectral Indices and Accuracy Assessment*

In the SPSS software (version 27.0), linear, quadratic, and cubic polynomial models (Equations (3)–(5)) were employed to analyze the relationship between the CBI (independent variable) and various spectral indices (dependent variables). The coefficient of determination ($R^2$) and the root mean square error (RMSE) were used to evaluate the fitting

capacity of the models on both the training and validation datasets. Table 4 lists the ten spectral indices used for assessing fire severity.

$$\text{Linear model: } y = a(x) + b \tag{3}$$

$$\text{Quadratic polynomial model: } y = a(x)^2 + b(x) + c \tag{4}$$

$$\text{Cubic polynomial model: } y = a(x)^3 + b(x)^2 + c(x) + d \tag{5}$$

**Table 4.** Spectral indices are used to evaluate burn severity.

| Spectral Index | Computational Formula | Reference |
|---|---|---|
| Analytical Burned Area Index (ABAI) | $\frac{3*SWIR2 - 2*SWIR1 - 3*GREEN}{3*SWIR2 + 2*SWIR1 + 3*GREEN}$ | [24] |
| Differenced Analytical Burned Area Index (dABAI) | $ABAIpre - ABAIpost$ | [24] |
| Normalized Burn Ratio (NBR) | $\frac{NIR - SWIR2}{NIR + SWIR2}$ | [23] |
| Differenced Normalized Burn Ratio (dNBR) | $NBRpre - NBRpost$ | [29] |
| Normalized Difference Vegetation Index (NDVI) | $\frac{NIR - RED}{NIR + RED}$ | [34] |
| Differenced Normalized Difference Vegetation Index (dNDVI) | $NDVIpre - NDVIpost$ | [34] |
| Soil-Adjusted Vegetation Index (SAVI) | $(1 + L)*\frac{NIR - RED}{NIR + RED + L}$, where L = 0.5 | [35] |
| Differenced Soil-Adjusted Vegetation Index (dSAVI) | $SAVIpre - SAVIpost$ | [35] |
| Mid-Infrared Bi-Spectral Index (MIRBI) | $10*SWIR2 - 9.8*SWIR1 + 2$ | [36] |
| Differenced Mid-Infrared Bi-Spectral Index (dMIRBI) | $MIRBIpre - MIRBIpost$ | [36] |
| Burned Area Index (BAI) | $\frac{1}{(0.1 - RED)^2 + (0.06 - NIR)^2}$ | [21] |
| Differenced Burned Area Index (dBAI) | $BAIpre - BAIpost$ | [21] |
| Char Soil Index (CSI) | $\frac{NIR}{SWIR2}$ | [22] |
| Differenced Char Soil Index (dCSI) | $CSIpre - CSIpost$ | [22] |

In the formulas, the independent variable (x) represents the CBI value, and the dependent variable (y) corresponds to the spectral index values of the respective CBI sampling points.

This study employed a confusion matrix and the metrics derived from it to evaluate the accuracy of various spectral indices in recognizing fire severity. The confusion matrix depicts the correspondence between the classification results of the spectral indices and the actual ground conditions. The assessment accuracy of each spectral index classification image is measured using the following metrics: Producer Accuracy (PA): PA represents the proportion of actual burned areas that are correctly classified as such by the spectral index. It measures the ability of the spectral index to accurately identify the true extent of burned areas. User Accuracy (UA): UA represents the proportion of burned areas correctly classified by the spectral index out of all the areas classified as burned. It measures the effectiveness of the spectral index in correctly identifying burned areas without many false positives. Overall accuracy (OA): OA is a metric that indicates the overall correctness of the spectral index in classifying fire severity. It provides a general measure of the spectral index's accuracy in classifying different fire severity levels. Kappa coefficient (kappa): The kappa coefficient is a statistical measure that takes into account the possibility of correct classification occurring by chance. It quantifies the level of agreement between the classification results of the spectral index and the actual conditions, providing a more robust assessment of classification performance. These metrics are commonly used for assessing the accuracy of classification results and are crucial for evaluating the performance of spectral indices in identifying different fire severity levels [37,38].

## 3. Results

### 3.1. Validation of the ABAI for Fire Severity Detection

To validate the effectiveness of the ABAI in assessing fire severity, the study area was assessed using Sentinel-2 remote sensing imagery data from Ganzhou City, Jiangxi Province, China. In the SPSS software, CBI sampling points for the two study areas were fitted to various spectral indices (as shown in Figure 4). The best-fitting model was selected to construct fire severity thresholds based on the $R^2$ and RMSE of each model's fitting results. Table 5 presents the fitting results for the three regression models.

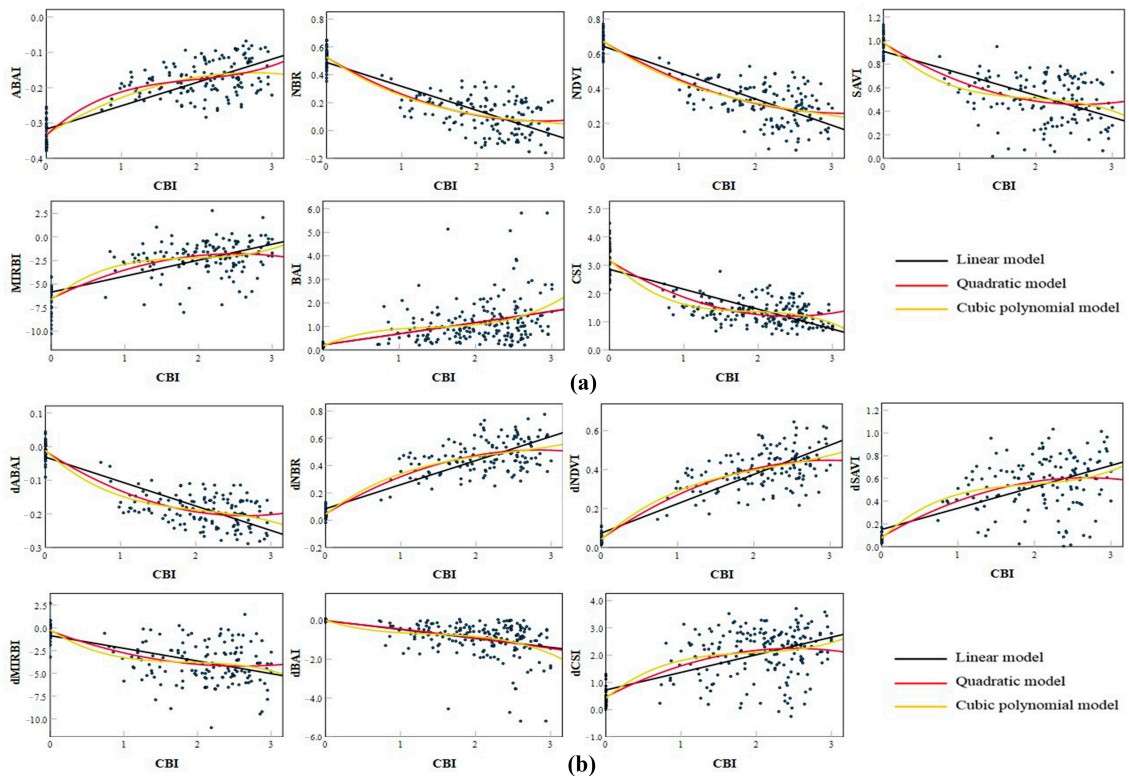

**Figure 4.** Scatterplots representing the model relationship between the CBI and spectral indices. (**a**) Uni-temporal remote sensing imagery data. (**b**) Bi-temporal remote sensing imagery data.

**Table 5.** Results of independent validation using the observed CBI and different burn severity indices in Ganzhou City, Jiangxi Province, China.

| Model Spectral Indices | Linear $R^2$ | RMSE | Quadratic Polynomial $R^2$ | RMSE | Cubic Polynomial $R^2$ | RMSE |
|---|---|---|---|---|---|---|
| NBR | **0.664** | **0.842** | 0.665 | 0.839 | 0.667 | 0.837 |
| NDVI | **0.642** | **0.921** | 0.645 | 0.916 | 0.645 | 0.914 |
| SAVI | **0.453** | **1.581** | 0.455 | 1.578 | 0.463 | 1.572 |
| MIRBI | **0.441** | **12.874** | 0.443 | 12.865 | 0.447 | 12.852 |
| BAI | **0.230** | **15.283** | 0.232 | 15.280 | 0.233 | 15.272 |
| CSI | **0.625** | **0.913** | 0.628 | 0.910 | 0.629 | 0.905 |
| ABAI | **0.658** | **0.782** | 0.662 | 0.779 | 0.664 | 0.778 |
| dNBR | **0.735** | **0.586** | 0.736 | 0.574 | 0.737 | 0.568 |
| dNDVI | **0.713** | **0.663** | 0.715 | 0.659 | 0.721 | 0.653 |
| dSAVI | **0.474** | **1.468** | 0.477 | 1.464 | 0.478 | 1.462 |
| dMIRBI | **0.290** | **12.765** | 0.293 | 12.748 | 0.299 | 12.741 |
| dBAI | **0.233** | **14.090** | 0.237 | 13.876 | 0.240 | 13.854 |
| dCSI | **0.407** | **2.037** | 0.409 | 1.987 | 0.410 | 1.985 |
| dABAI | **0.728** | **0.448** | 0.731 | 0.433 | 0.733 | 0.432 |

By comparing the R-squared (R$^2$) and root mean square error (RMSE) of the regression models, we found that the three models had similar fitting results for the regression relationship between spectral indices and fire severity in Ganzhou City, Jiangxi Province, China. Ultimately, the linear model was selected to establish the threshold for fire severity. Table 6 presents the fire severity thresholds for various spectral indices, and Figures 5 and 6 illustrate fire severity maps for various spectral indices in the study area using Sentinel-2 uni-temporal and bi-temporal imagery data.

**Table 6.** Fire severity thresholds for each spectral index in Ganzhou City, Jiangxi Province, China.

| Spectral Indices | Severity Grade | | | |
|---|---|---|---|---|
| | Unburned | Low | Moderate | High |
| NBR | >0.425 | 0.263~0.425 | 0.102~0.263 | <0.102 |
| NDVI | >0.593 | 0.451~0.593 | 0.310~0.451 | <0.310 |
| SAVI | >0.888 | 0.681~0.888 | 0.475~0.681 | <0.475 |
| MIRBI | <−5.573 | −5.573~−3.747 | −3.747~−1.921 | >−1.921 |
| BAI | <0.360 | 0.360~0.825 | 0.825~1.289 | >1.289 |
| CSI | >2.654 | 1.951~2.654 | 1.249~1.951 | <1.249 |
| ABAI | <−0.294 | −0.294~−0.230 | −0.230~−0.167 | >−0.167 |
| dNBR | <0.134 | 0.134~0.287 | 0.287~0.439 | >0.439 |
| dNDVI | <0.120 | 0.120~0.249 | 0.249~0.378 | >0.378 |
| dSAVI | <0.183 | 0.183~0.381 | 0.381~0.579 | >0.579 |
| dMIRBI | >−1.073 | −2.624~−1.073 | −4.175~−2.624 | <−4.175 |
| dBAI | >−0.167 | −0.614~−0.167 | −1.061~−0.614 | <−1.061 |
| dCSI | <0.908 | 0.908~1.546 | 1.546~2.183 | >2.183 |
| dABAI | >−0.049 | −0.111~−0.049 | −0.174~−0.111 | <−0.174 |

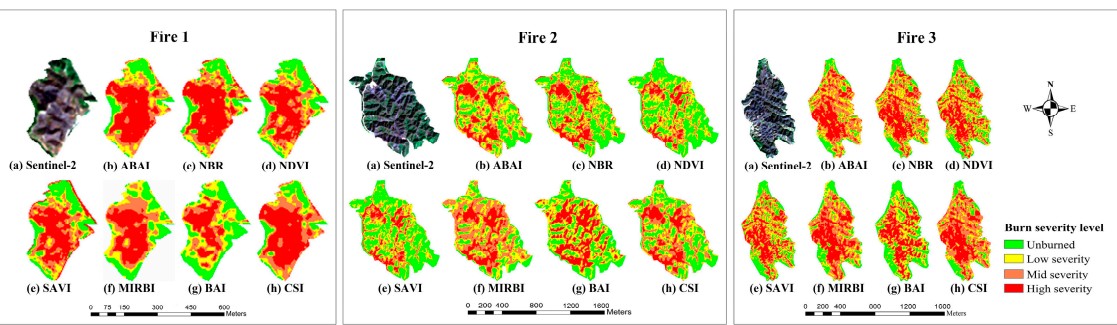

**Figure 5.** Sentinel-2 uni-temporal image for mapping the severity of burns in Ganzhou City, Jiangxi Province, China.

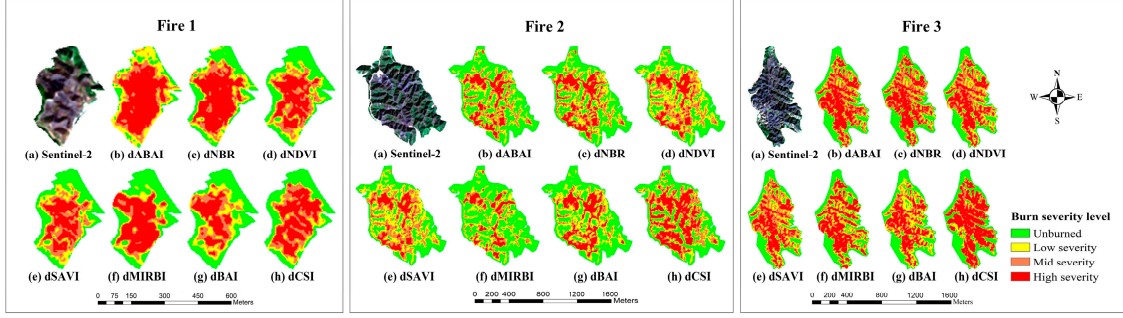

**Figure 6.** Sentinel-2 vi-temporal image for mapping the severity of burns in Ganzhou City, Jiangxi Province, China.

Table 7 provides confusion matrices for the fire severity classification results from the seven spectral indices based on Sentinel-2 imagery data. The values in the confusion matrix

represent the evaluation metrics' mean and variance over five repetitions of the experiment. From the results in the confusion matrix, the ABAI shows high overall accuracy and kappa coefficient values, ranking first among all spectral indices. Compared with the NBR, the ABAI exhibits a significant advantage in classifying unburned and highly burned severity levels. According to the PA and UA of the ABAI for low and medium fire severity, it can be seen that the main factor affecting the overall accuracy of the ABAI is confusion between low and medium fire severity, which is the same as that of the NBR. The dABAI also has the same problem, although the overall accuracy is improved compared with that of the ABAI; the main factor affecting the overall accuracy is still confusion between low and medium fire severity. Additionally, when comparing the classification accuracy of fire severity for various spectral indices, it is evident that the ABAI maintains good stability throughout the repeated experiments, with most accuracy evaluation metrics showing minimal variance. This indicates that the ABAI can effectively assess fire severity using Sentinel-2 imagery and offers better classification accuracy and stability compared with other spectral indices.

### 3.2. Assessment of the Impacts of Different Sensors

To further investigate the impacts of sensors on the classification accuracy of fire severity using the ABAI, assessments were carried out using Landsat 8 OLI data for Ganzhou City, Jiangxi Province, China. Similar to the previous approach, CBI sampling points were fitted to various spectral indices, as shown in Figure 7. The best-fitting model was selected based on the $R^2$ and RMSE of each model's fitting results, and Table 8 provides fitting results for the three regression models.

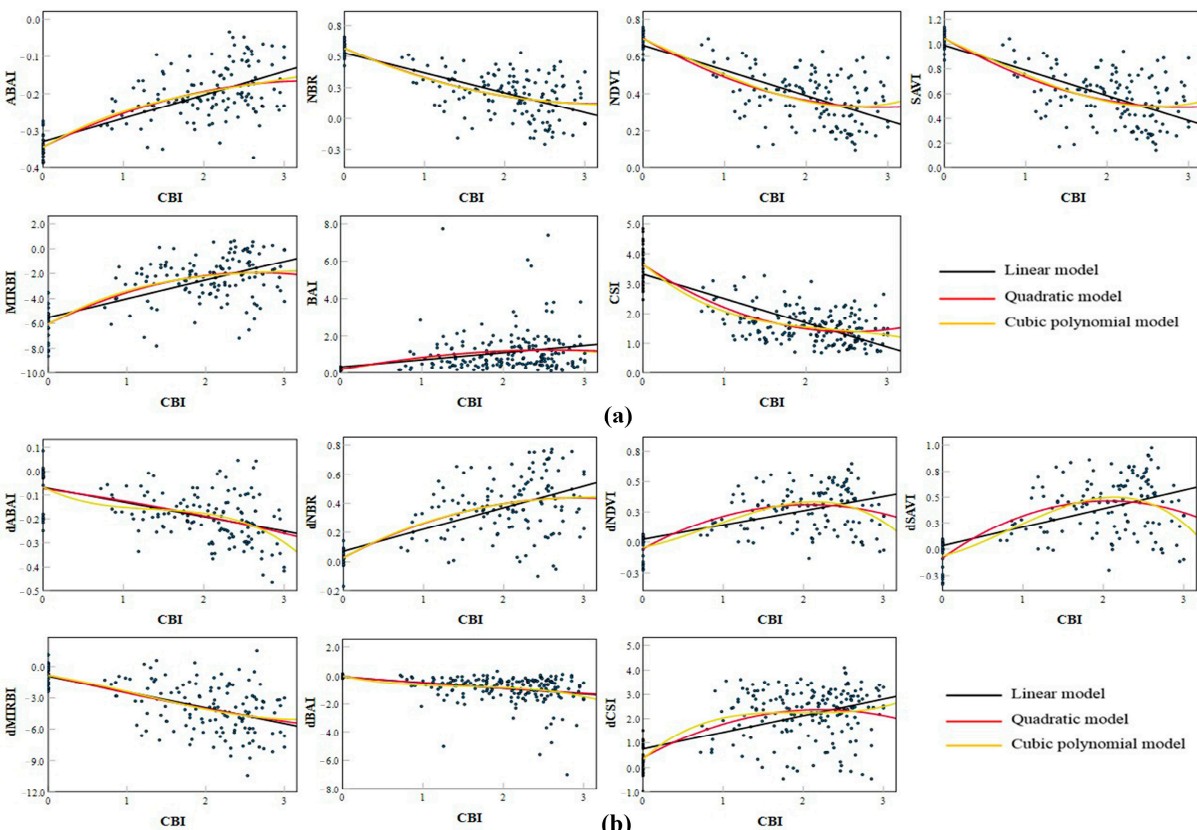

**Figure 7.** Scatterplots representing the model relationship between the CBI and spectral indices. (**a**) Uni-temporal remote sensing imagery data. (**b**) Bi-temporal remote sensing imagery data.

**Table 7.** Confusion matrix and accuracy assessment of burn severity derived from Ganzhou City, Jiangxi Province, China (Note that values with the highest accuracy are highlighted in bold).

| Severity Grade | | Burn Index | | | | | | |
|---|---|---|---|---|---|---|---|---|
| | | NBR | NDVI | SAVI | MIRBI | BAI | CSI | ABAI |
| Producer Accuracy (%) | Unburned | 88.34 ± 12.64 | 70.00 ± 9.50 | 83.33 ± 10.21 | 80.00 ± 12.64 | 86.37 ± 6.43 | 86.37 ± 6.43 | **90.00 ± 6.97** |
| | Low | **53.33 ± 7.46** | 20.00 ± 13.94 | 46.67 ± 21.73 | 6.67 ± 9.13 | 50.00 ± 23.57 | 50.00 ± 23.58 | 46.67 ± 13.95 |
| | Moderate | 37.04 ± 13.61 | 30.37 ± 4.83 | 34.07 ± 11.54 | **48.15 ± 6.93** | 19.23 ± 5.44 | 42.31 ± 16.32 | 39.26 ± 10.00 |
| | High | 55.35 ± 14.92 | **60.60 ± 6.98** | 51.11 ± 6.09 | 54.07 ± 8.53 | 38.00 ± 2.83 | 46.00 ± 8.49 | 56.13 ± 4.95 |
| User Accuracy (%) | Unburned | 80.34 ± 13.42 | 62.82 ± 14.27 | **82.44 ± 6.45** | 80.24 ± 10.36 | 57.57 ± 6.98 | 76.19 ± 33.67 | 81.57 ± 11.20 |
| | Low | **32.89 ± 8.54** | 12.80 ± 9.51 | 17.81 ± 8.43 | 4.40 ± 6.46 | 14.36 ± 5.41 | 24.29 ± 6.06 | 29.14 ± 19.05 |
| | Moderate | 46.76 ± 7.70 | 46.08 ± 5.71 | 41.91 ± 8.99 | 49.71 ± 6.80 | 41.26 ± 6.92 | **54.91 ± 9.38** | 49.90 ± 3.80 |
| | High | 56.63 ± 6.26 | 55.87 ± 5.74 | 63.65 ± 11.93 | 59.13 ± 6.59 | 53.13 ± 4.42 | **63.98 ± 10.54** | 62.62 ± 13.67 |
| Overall Accuracy (%) | | 53.78 ± 2.64 | 47.33 ± 3.56 | 49.72 ± 3.86 | 52.22 ± 4.87 | 40.45 ± 1.04 | 52.21 ± 5.20 | **54.62 ± 2.13** |
| Kappa Coefficient | | 0.35 ± 0.02 | 0.26 ± 0.04 | 0.31 ± 0.06 | 0.31 ± 0.07 | 0.23 ± 0.01 | 0.35 ± 0.01 | **0.37 ± 0.02** |
| | | dNBR | dNDVI | dSAVI | dMIRBI | dBAI | dCSI | dABAI |
| Producer Accuracy (%) | Unburned | 91.67 ± 11.79 | 80.00 ± 12.64 | **100.00 ± 0.00** | 85.00 ± 6.97 | 95.46 ± 6.43 | 90.91 ± 0.00 | 90.00 ± 10.87 |
| | Low | **66.67 ± 27.39** | 50.00 ± 11.79 | 33.33 ± 16.67 | 36.67 ± 18.26 | 58.34 ± 11.79 | 25.00 ± 35.36 | 63.33 ± 16.67 |
| | Moderate | 40.74 ± 10.14 | 40.74 ± 11.71 | 38.52 ± 8.53 | 24.44 ± 6.20 | 36.49 ± 13.67 | 21.15 ± 8.16 | **45.19 ± 7.59** |
| | High | 55.30 ± 8.31 | 53.02 ± 12.94 | 50.37 ± 10.67 | 48.89 ± 12.67 | 54.00 ± 8.49 | 56.00 ± 5.66 | **59.03 ± 7.70** |
| User Accuracy (%) | Unburned | 82.25 ± 9.37 | 83.32 ± 6.54 | 68.33 ± 8.98 | 48.36 ± 4.28 | 74.02 ± 13.17 | 48.06 ± 6.47 | **84.84 ± 6.07** |
| | Low | 30.91 ± 11.31 | 25.35 ± 15.05 | 20.91 ± 14.09 | 15.40 ± 6.85 | 32.90 ± 24.19 | 25.00 ± 35.36 | **35.21 ± 12.12** |
| | Moderate | 53.19 ± 3.16 | 49.29 ± 3.89 | 49.61 ± 3.98 | 43.92 ± 8.89 | 54.17 ± 5.89 | 49.15 ± 6.65 | **56.20 ± 6.39** |
| | High | 61.12 ± 5.26 | 62.32 ± 9.80 | 62.77 ± 4.21 | 60.16 ± 9.45 | 61.25 ± 1.77 | 49.08 ± 1.31 | **64.50 ± 5.75** |
| Overall Accuracy (%) | | 56.58 ± 1.62 | 52.65 ± 3.82 | 52.78 ± 4.05 | 44.72 ± 4.10 | 54.41 ± 8.32 | 45.59 ± 2.08 | **59.66 ± 2.07** |
| Kappa Coefficient | | 0.40 ± 0.02 | 0.34 ± 0.05 | 0.35 ± 0.05 | 0.26 ± 0.05 | 0.38 ± 0.08 | 0.25 ± 0.03 | **0.43 ± 0.03** |

**Table 8.** Results of independent validation using the observed CBI and different burn severity indices in Ganzhou City, Jiangxi Province, China.

| Model Spectral Indices | Linear $R^2$ | RMSE | Quadratic Polynomial $R^2$ | RMSE | Cubic Polynomial $R^2$ | RMSE |
|---|---|---|---|---|---|---|
| NBR | **0.489** | **0.856** | 0.490 | 0.854 | 0.492 | 0.854 |
| NDVI | **0.485** | **1.070** | 0.486 | 1.069 | 0.489 | 1.068 |
| SAVI | **0.435** | **1.755** | 0.438 | 1.754 | 0.439 | 1.754 |
| MIRBI | **0.405** | **12.369** | 0.408 | 12.363 | 0.410 | 12.359 |
| BAI | **0.120** | **10.658** | 0.120 | 10.651 | 0.122 | 10.586 |
| CSI | **0.524** | **1.267** | 0.526 | 1.263 | 0.526 | 1.261 |
| ABAI | **0.487** | **0.733** | 0.489 | 0.730 | 0.490 | 0.729 |
| dNBR | **0.497** | **0.737** | 0.498 | 0.735 | 0.499 | 0.732 |
| dNDVI | **0.492** | **0.953** | 0.493 | 0.951 | 0.494 | 0.950 |
| dSAVI | **0.442** | **1.773** | 0.444 | 1.772 | 0.445 | 1.768 |
| dMIRBI | **0.326** | **12.703** | 0.328 | 12.701 | 0.329 | 12.665 |
| dBAI | **0.150** | **15.763** | 0.150 | 15.747 | 0.156 | 15.744 |
| dCSI | **0.365** | **9.879** | 0.367 | 9.872 | 0.369 | 9.865 |
| dABAI | **0.493** | **0.716** | 0.495 | 0.714 | 0.496 | 0.713 |

Consistent with the results from the Sentinel-2 imagery experiment, the linear model was selected to construct fire severity thresholds based on the model's outcomes. Table 9 provides the fire severity thresholds for various spectral indices, and Figures 8 and 9 depict fire severity maps for various spectral indices in the study area using Landsat 8 OLI uni-temporal and bi-temporal imagery data.

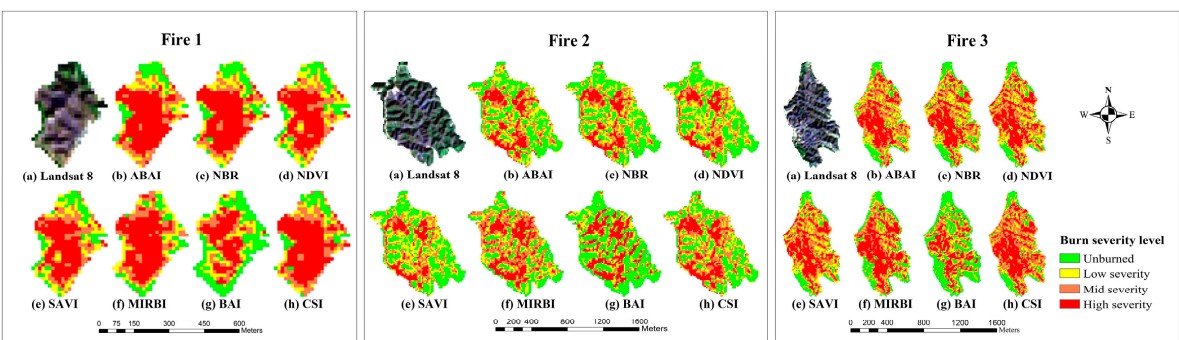

**Figure 8.** Landsat 8 OLI uni-temporal image for mapping the severity of burns in Ganzhou City, Jiangxi Province, China.

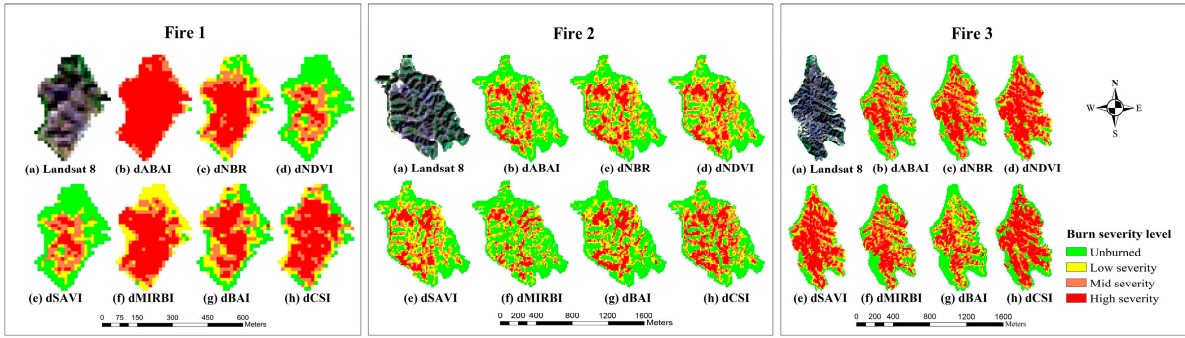

**Figure 9.** Landsat 8 OLI bi-temporal image for mapping the severity of burns in Ganzhou City, Jiangxi Province, China.

**Table 9.** Fire severity thresholds for each spectral index in Ganzhou City, Jiangxi Province, China.

| Spectral Indices | Severity Grade | | | |
|---|---|---|---|---|
| | Unburned | Low | Moderate | High |
| NBR | >0.474 | 0.318~0.474 | 0.161~0.318 | <0.161 |
| NDVI | >0.618 | 0.483~0.618 | 0.348~0.483 | <0.348 |
| SAVI | >0.927 | 0.724~0.927 | 0.522~0.724 | <0.522 |
| MIRBI | <−5.171 | −5.171~−3.630 | −3.630~−2.089 | >−2.089 |
| BAI | <0.436 | 0.436~0.806 | 0.806~1.176 | >1.176 |
| CSI | >3.041 | 2.279~3.041 | 1.484~2.279 | <1.484 |
| ABAI | <−0.307 | −0.307~−0.247 | −0.247~−0.187 | >−0.187 |
| dNBR | <0.114 | 0.114~0.260 | 0.260~0.405 | >0.405 |
| dNDVI | <0.067 | 0.067~0.182 | 0.182~0.297 | >0.297 |
| dSAVI | <0.100 | 0.100~0.273 | 0.273~0.445 | >0.445 |
| dMIRBI | >−1.364 | −2.882~−1.364 | −4.399~−2.882 | <−4.399 |
| dBAI | >−0.193 | −0.592~−0.193 | −0.991~−0.592 | <−0.991 |
| dCSI | <0.978 | 0.978~1.644 | 1.644~2.310 | >2.310 |
| dABAI | >−0.078 | −0.139~−0.078 | −0.200~−0.139 | <−0.200 |

Table 10 presents a confusion matrix for the fire severity classification results using Landsat 8 OLI data from Ganzhou City, Jiangxi Province, China. Comparing the results from the confusion matrix with the Sentinel-2 experiment, although the ABAI exhibits a slight decrease in overall accuracy and kappa coefficient for uni-temporal fire severity classification, it still maintains the highest accuracy among the spectral indices and remains stable. Furthermore, most accuracy metrics are higher than those for other spectral indices. Regarding the fire severity classification results using bi-temporal data, the dABAI's overall accuracy and kappa coefficient rank second, slightly lower than the dNBR. However, it still demonstrates significant advantages and stability in classifying unburned and highly burned areas. This suggests that the ABAI also performs well in assessing fire severity using Landsat 8 satellite data, with good classification accuracy and stability. It exhibits a significant advantage in assessing fire severity in areas with complex terrain and mountain shadow zones. In addition, a comparison of the PA and UA for each fire severity level of the ABAI shows that, unlike the use of Sentinel-2 data, the main reason for the overall accuracy of the ABAI being affected by the use of Landsat 8 data is the misclassification of unburned fire severity versus low fire severity. In the case of the dABAI, as with the use of Sentinel-2 data, the main reason for the overall accuracy is still the misclassification of low and moderate fire severity.

*3.3. Assessment of Impacts on Different Regions*

To assess the potential of the ABAI to evaluate fire severity in different regions, the study area in Okanogan, Washington, USA, was evaluated using Landsat 8 OLI data, similar to the experiments described above. In the SPSS software, the CBI sampling points for the two study areas were fitted to various spectral indices (as shown in Figure 10). The best-fitting model was selected for constructing the fire severity thresholds based on the $R^2$ and RMSE of each model's fitting results. Table 11 presents the fitting results for the three regression models.

**Table 10.** Confusion matrix and accuracy assessment of burn severity derived from using Landsat 8 images of Ganzhou City, Jiangxi Province, China. (Note that values with the highest accuracy are highlighted in bold.)

| Severity Grade | | Burn Index | | | | | | |
|---|---|---|---|---|---|---|---|---|
| | | **NBR** | **NDVI** | **SAVI** | **MIRBI** | **BAI** | **CSI** | **ABAI** |
| Producer Accuracy (%) | Unburned | 90.91 ± 9.09 | 87.27 ± 13.79 | 83.64 ± 14.94 | 76.37 ± 13.79 | **96.97 ± 5.25** | 78.79 ± 10.50 | 85.46 ± 8.13 |
| | Low | 20.00 ± 7.45 | **43.33 ± 9.13** | 36.67 ± 13.94 | 33.33 ± 11.78 | 22.22 ± 9.62 | 27.78 ± 9.62 | 16.67 ± 11.78 |
| | Moderate | 31.75 ± 6.14 | 15.51 ± 4.72 | 20.92 ± 8.83 | 27.97 ± 10.91 | 7.85 ± 4.08 | 34.05 ± 19.01 | **35.63 ± 6.12** |
| | High | 57.60 ± 6.07 | 57.60 ± 6.69 | 57.60 ± 8.29 | 57.60 ± 6.07 | 40.56 ± 10.43 | **61.66 ± 13.71** | 59.20 ± 5.22 |
| User Accuracy (%) | Unburned | **67.69 ± 15.86** | 54.65 ± 2.65 | 51.54 ± 6.95 | 59.03 ± 18.53 | 46.71 ± 7.89 | 59.15 ± 20.94 | 61.14 ± 11.51 |
| | Low | 13.54 ± 5.87 | 22.69 ± 9.20 | 22.55 ± 10.61 | 18.58 ± 0.37 | 6.01 ± 1.08 | **30.00 ± 8.66** | 10.79 ± 7.23 |
| | Moderate | 53.79 ± 6.81 | 51.00 ± 16.75 | 54.09 ± 16.63 | 51.61 ± 8.86 | 31.39 ± 12.81 | **60.54 ± 10.62** | 58.70 ± 6.92 |
| | High | 52.93 ± 3.73 | 48.04 ± ±5.18 | 49.17 ± 6.29 | 50.46 ± 6.12 | 50.42 ± 12.54 | 51.67 ± 2.89 | **53.72 ± 3.81** |
| Overall Accuracy (%) | | 49.86 ± 2.88 | 45.75 ± 2.10 | 46.64 ± 3.20 | 47.22 ± 5.02 | 35.53 ± 2.86 | 47.47 ± 1.75 | **50.75 ± 2.86** |
| Kappa coefficient | | 0.30 ± 0.03 | 0.25 ± 0.04 | 0.26 ± 0.05 | 0.26 ± 0.06 | 0.17 ± 0.04 | 0.30 ± 0.02 | **0.31 ± 0.05** |
| | | dNBR | dNDVI | dSAVI | dMIRBI | dBAI | dCSI | dABAI |
| Producer Accuracy (%) | Unburned | 87.27 ± 8.13 | 94.55 ± 4.98 | 89.09 ± 7.61 | 69.09 ± 15.21 | **96.97 ± 5.25** | 84.85 ± 10.50 | 90.91 ± 6.43 |
| | Low | **50.00 ± 16.67** | 40.00 ± 9.13 | 40.00 ± 14.91 | 30.00 ± 21.73 | 33.33 ± 9.62 | 38.89 ± 9.62 | 40.00 ± 14.91 |
| | Moderate | **38.00 ± 8.83** | 33.39 ± 6.78 | 36.68 ± 18.63 | 23.38 ± 11.75 | 15.59 ± 3.86 | 33.80 ± 9.80 | 31.84 ± 5.57 |
| | High | 60.00 ± 5.66 | 55.20 ± 9.55 | 55.20 ± 6.57 | 52.80 ± 4.38 | 42.02 ± 16.75 | 59.97 ± 8.90 | **62.40 ± 3.58** |
| User Accuracy (%) | Unburned | **82.73 ± 5.95** | 81.59 ± 7.73 | 79.22 ± 14.66 | 52.39 ± 15.31 | 60.27 ± 4.91 | 51.75 ± 4.32 | 82.05 ± 6.55 |
| | Low | 25.32 ± 8.78 | 22.61 ± 6.09 | 22.14 ± 7.23 | 17.58 ± 7.97 | 9.68 ± 0.28 | **27.86 ± 2.58** | 19.92 ± 9.30 |
| | Moderate | 56.56 ± 4.06 | 49.35 ± 12.49 | 51.14 ± 9.60 | 38.83 ± 13.02 | 44.29 ± 5.15 | **68.22 ± 1.36** | 54.94 ± 9.17 |
| | High | **56.03 ± 4.61** | 52.73 ± 4.22 | 54.50 ± 5.94 | 47.77 ± 5.26 | 52.62 ± 12.94 | 52.17 ± 3.76 | 55.80 ± 4.28 |
| Overall Accuracy (%) | | **55.18 ± 4.73** | 51.93 ± 3.62 | 52.24 ± 5.11 | 42.21 ± 6.86 | 40.53 ± 4.46 | 51.97 ± 3.10 | 53.42 ± 5.62 |
| Kappa Coefficient | | **0.37 ± 0.06** | 0.33 ± 0.05 | 0.33 ± 0.06 | 0.21 ± 0.10 | 0.24 ± 0.06 | 0.33 ± 0.04 | 0.35 ± 0.08 |

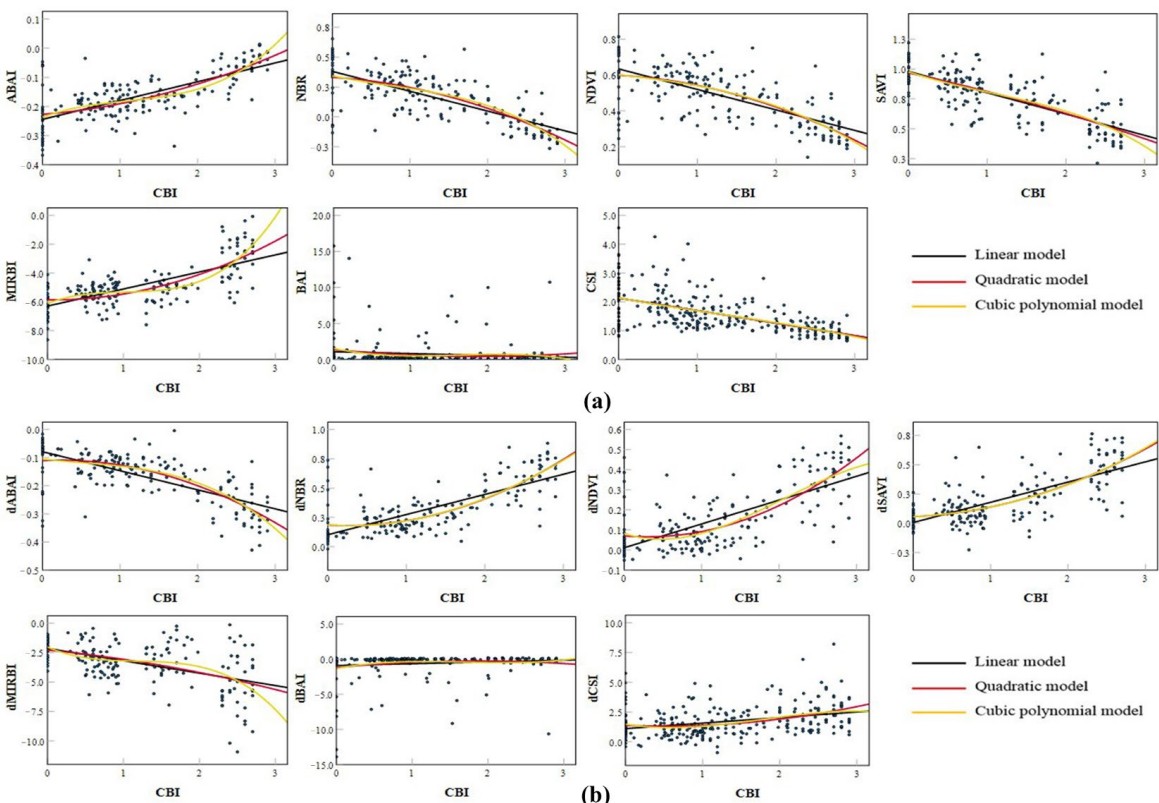

**Figure 10.** Scatterplots representing the model relationship between the CBI and spectral indices. (**a**) Uni-temporal remote sensing imagery data. (**b**) Bi-temporal remote sensing imagery data.

**Table 11.** Results of independent validation using the observed CBI and different burn severity indices in Okanogan County, Washington, USA.

| Model Spectral Indices | Linear $R^2$ | RMSE | Quadratic Polynomial $R^2$ | RMSE | Cubic Polynomial $R^2$ | RMSE |
|---|---|---|---|---|---|---|
| NBR | 0.522 | 2.043 | 0.548 | 1.480 | **0.552** | **1.213** |
| NDVI | 0.466 | 1.409 | 0.491 | 1.023 | **0.491** | **0.835** |
| SAVI | 0.564 | 2.134 | 0.565 | 1.512 | **0.568** | **1.237** |
| MIRBI | 0.488 | 14.181 | 0.533 | 10.475 | **0.569** | **8.839** |
| BAI | 0.118 | 3.869 | 0.155 | 3.594 | **0.181** | **3.421** |
| CSI | 0.335 | 6.450 | 0.335 | 4.561 | **0.335** | **3.725** |
| ABAI | 0.519 | 0.796 | 0.539 | 0.574 | **0.556** | **0.476** |
| dNBR | 0.537 | 2.120 | 0.605 | 1.591 | **0.605** | **1.299** |
| dNDVI | 0.485 | 1.463 | 0.552 | 1.104 | **0.559** | **0.907** |
| dSAVI | 0.520 | 2.067 | 0.550 | 1.503 | **0.550** | **1.227** |
| dMIRBI | 0.244 | 12.621 | 0.247 | 8.987 | **0.271** | **7.675** |
| dBAI | 0.126 | 3.732 | 0.168 | 3.511 | **0.187** | **3.190** |
| dCSI | 0.328 | 6.948 | 0.360 | 5.387 | **0.370** | **4.520** |
| dABAI | 0.494 | 0.835 | 0.554 | 0.625 | **0.559** | **0.513** |

Based on the $R^2$ and RMSE of the fitting results for each regression model, the cubic polynomial model was selected for Okanogan County, Washington. Table 12 provides the fire severity thresholds for various spectral indices, and Figures 11 and 12 depict fire severity maps for various spectral indices in the study area using Landsat 8 OLI uni-temporal and bi-temporal imagery data.

**Table 12.** Fire severity thresholds for each spectral index in Okanogan County, Washington, USA.

| Spectral Indices | Severity Grade | | | |
|---|---|---|---|---|
| | Unburned | Low | Moderate | High |
| NBR | >0.316 | 0.207~0.316 | 0.026~0.207 | <0.026 |
| NDVI | >0.590 | 0.514~0.590 | 0.378~0.514 | <0.378 |
| SAVI | >0.914 | 0.780~0.914 | 0.622~0.780 | <0.622 |
| MIRBI | <−5.573 | −5.573~−3.747 | −3.747~−1.921 | >−1.921 |
| BAI | >1.196 | 0.701~1.196 | 0.496~0.701 | <0.496 |
| CSI | >2.070 | 1.594~2.070 | 1.183~1.594 | <1.183 |
| ABAI | <−0.294 | −0.294~−0.230 | −0.230~−0.167 | >−0.167 |
| dNBR | <0.176 | 0.176~0.257 | 0.257~0.487 | >0.487 |
| dNDVI | <0.057 | 0.057~0.116 | 0.116~0.296 | >0.296 |
| dSAVI | <0.071 | 0.071~0.173 | 0.173~0.410 | >0.410 |
| dMIRBI | >−2.696 | −3.217~−2.696 | −4.095~−3.217 | <−4.095 |
| dBAI | <−1.047 | −1.047~−0.561 | −0.561~−0.356 | >−0.356 |
| dCSI | <1.234 | 1.234~1.444 | 1.444~2.333 | >2.333 |
| dABAI | >−0.110 | −0.141~−0.110 | −0.220~−0.141 | <−0.220 |

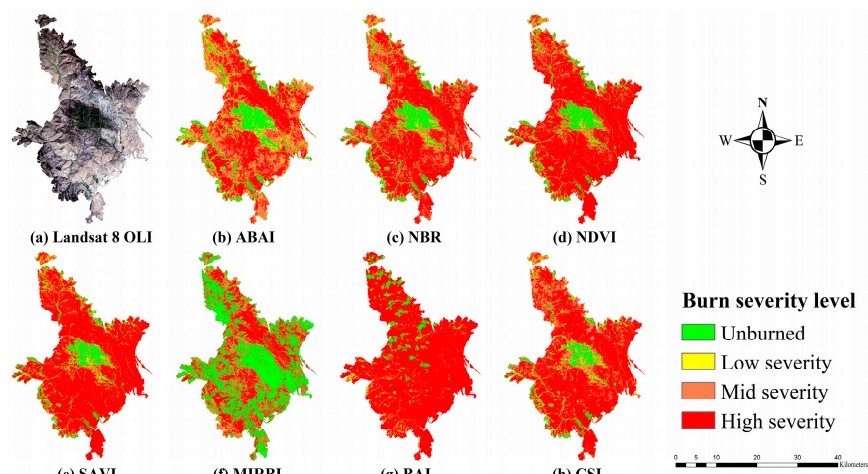

**Figure 11.** Landsat 8 uni-temporal image for mapping the severity of burns in Okanogan County, Washington, USA.

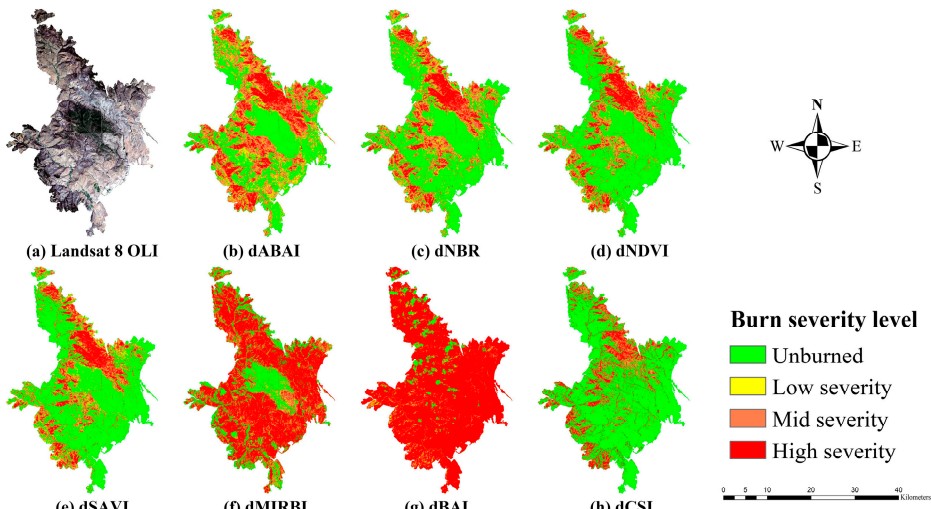

**Figure 12.** Landsat 8 bi-temporal image for mapping the severity of burns in Okanogan County, Washington, USA.

Table 13 presents a confusion matrix for the classification results for fire severity using Landsat 8 OLI data for seven spectral indices. Based on the results from the confusion matrix, the ABAI spectral index continues to exhibit high overall accuracy and kappa coefficients, ranking among the top two indices. Compared with the NBR, it maintains a significant advantage in the classification of unburned and highly burned areas. Additionally, when comparing the classification accuracy of fire severity for different spectral indices, it is obvious that the ABAI exhibits good stability across repeated experiments, with most accuracy evaluation metrics displaying minimal variance. This suggests that, compared with other spectral indices, the ABAI remains effective in assessing fire severity at high latitudes and maintains good sensitivity in evaluating fire severity. In addition to this, by comparing the PA and UA for each fire severity class of the ABAI in the high-latitude study area, it can be seen that, similar to the results for the low-latitude study area, the main factor affecting the overall accuracy of the ABAI is still the misclassification of unburned fire severity versus low fire severity. In contrast, the main factor affecting the overall accuracy of the dABAI is the misclassification of unburned fire severity versus low fire severity and low and moderate fire severity.

**Table 13.** Confusion matrix and accuracy assessment of burn severity derived using Landsat 8 images of Okanogan County, Washington, USA. (Note that values with the highest accuracy are highlighted in bold.)

| Severity Grade | | Burn Index | | | | | | |
|---|---|---|---|---|---|---|---|---|
| | | **NBR** | **NDVI** | **SAVI** | **MIRBI** | **BAI** | **CSI** | **ABAI** |
| Producer Accuracy (%) | Unburned | **83.08 ± 10.03** | **83.08 ± 10.03** | 44.62 ± 23.33 | 44.62 ± 19.15 | 10.25 ± 4.44 | 38.46 ± 7.69 | **83.08 ± 10.03** |
| | Low | 33.33 ± 15.63 | 30.67 ± 13.62 | **40.67 ± 8.94** | 14.67 ± 9.60 | 3.45 ± 3.45 | 24.14 ± 9.12 | 34.00 ± 9.54 |
| | Moderate | 42.50 ± 5.23 | 28.75 ± 9.48 | 33.75 ± 13.69 | 37.50 ± 8.84 | 0.00 ± 0.00 | **43.75 ± 3.61** | 28.75 ± 9.48 |
| | High | 84.44 ± 4.65 | 80.00 ± 3.04 | **97.65 ± 5.26** | 89.41 ± 10.52 | 90.74 ± 8.49 | 81.48 ± 6.41 | 87.78 ± 4.65 |
| User Accuracy (%) | Unburned | 52.82 ± 11.22 | 42.65 ± 2.00 | 42.86 ± 7.45 | 31.25 ± 5.04 | 35.32 ± 27.67 | 42.85 ± 17.16 | **56.99 ± 4.96** |
| | Low | 60.59 ± 17.98 | 60.13 ± 11.35 | **71.17 ± 3.11** | 54.58 ± 35.66 | 17.67 ± 16.62 | 58.25 ± 5.56 | 54.08 ± 1.38 |
| | Moderate | **43.86 ± 10.08** | 30.73 ± 8.97 | 42.62 ± 5.17 | 22.62 ± 3.37 | 0.00 ± 0.00 | 27.64 ± 0.62 | 28.65 ± 6.42 |
| | High | 65.50 ± 10.69 | 66.15 ± 2.40 | 53.84 ± 12.76 | 62.48 ± 6.26 | 24.47 ± 0.79 | 56.71 ± 5.87 | **69.42 ± 4.32** |
| Overall Accuracy (%) | | **55.59 ± 3.94** | 50.65 ± 5.35 | 52.63 ± 3.84 | 41.32 ± 2.73 | 24.56 ± 2.01 | 44.30 ± 1.52 | 53.77 ± 1.48 |
| Kappa coefficient | | **0.41 ± 0.04** | 0.35 ± 0.06 | 0.37 ± 0.06 | 0.24 ± 0.03 | 0.02 ± 0.02 | 0.27 ± 0.02 | 0.39 ± 0.01 |
| | | dNBR | dNDVI | dSAVI | dMIRBI | dBAI | dCSI | dABAI |
| Producer Accuracy (%) | Unburned | **87.69 ± 6.88** | 70.77 ± 11.41 | 52.11 ± 10.19 | 63.08 ± 12.64 | 10.25 ± 4.44 | 61.54 ± 0.00 | 86.16 ± 10.03 |
| | Low | **39.33 ± 10.90** | 31.33 ± 8.03 | 34.67 ± 6.91 | 14.00 ± 4.94 | 2.30 ± 1.99 | 9.22 ± 5.22 | 30.67 ± 14.61 |
| | Moderate | **53.75 ± 11.35** | 50.00 ± 13.26 | 33.75 ± 10.46 | 6.25 ± 4.42 | 4.17 ± 3.61 | 16.67 ± 7.22 | 43.75 ± 11.69 |
| | High | 62.22 ± 9.94 | 64.45 ± 12.79 | 72.94 ± 7.89 | **94.12 ± 5.88** | 85.18 ± 13.98 | 51.85 ± 21.03 | 72.22 ± 7.86 |
| User Accuracy (%) | Unburned | **48.92 ± 7.30** | 36.26 ± 3.09 | 33.18 ± 2.45 | 30.59 ± 3.34 | 33.06 ± 29.35 | 22.26 ± 1.05 | 47.54 ± 4.25 |
| | Low | **78.46 ± 5.62** | 59.33 ± 9.25 | 71.81 ± 6.09 | 67.26 ± 19.33 | 50.00 ± 30.83 | 44.44 ± 9.62 | 70.63 ± 2.56 |
| | Moderate | 39.19 ± 4.58 | **42.80 ± 6.91** | 38.02 ± 11.73 | 10.58 ± 6.59 | 13.33 ± 11.55 | 21.35 ± 3.20 | 32.40 ± 4.54 |
| | High | 69.32 ± 5.73 | **69.87 ± 8.36** | 51.35 ± 9.45 | 51.13 ± 9.75 | 23.61 ± 1.67 | 43.47 ± 3.95 | 69.15 ± 5.34 |
| Overall Accuracy (%) | | **55.84 ± 3.79** | 49.61 ± 1.09 | 46.05 ± 6.03 | 38.68 ± 2.73 | 23.68 ± 3.95 | 29.82 ± 2.01 | 52.47 ± 3.39 |
| Kappa Coefficient | | **0.42 ± 0.05** | 0.34 ± 0.02 | 0.29 ± 0.08 | 0.22 ± 0.04 | 0.04 ± 0.02 | 0.11 ± 0.03 | 0.38 ± 0.04 |

## 4. Discussion

In this study, based on Sentinel-2 and Landsat 8 OLI satellite data, combined with the CBI dataset, the ability of the ABAI and the six other spectral indices to assess fire severity in different regions was evaluated. According to the results of the fire severity assessment of Ganzhou City, Jiangxi Province, China, it was found that the ABAI consistently achieved the highest classification accuracy and stability in most cases when assessing fire severity using both uni-temporal and bi-temporal imagery from the Sentinel-2 satellite. This indicates that the ABAI is an effective method for assessing fire severity. In the fire experiment conducted in Washington State, USA, using Landsat 8 OLI satellite data, the ABAI's overall accuracy was very close to that of the Normalized Burn Ratio (NBR) in both uni-temporal and bi-temporal data and ranked second among all the spectral indices. This difference may be attributed to the varying central wavelengths of the bands for different satellite sensors [39,40]. For example, the SWIR2 band of Landsat 8 covers a wavelength window of 2107–2294 nm, whereas Sentinel-2 has a wavelength of 2202.4 nm; the SWIR1 band

of Landsat 8 covers a wavelength window of 1560–1660 nm, whereas Sentinel-2 has a wavelength of 1613.7 nm; and the green band of Landsat 8 covers a wavelength window of 525–600 nm, whereas Sentinel-2 has a wavelength of 559.8 nm. The ABAI was specifically designed using Sentinel-2 satellite data [24], which allows it to perform better in assessing fire severity when applied to Sentinel-2 imagery, as observed in this study.

Comparing the experimental results based on Landsat 8 OLI satellite data in Ganzhou City, Jiangxi Province, China, it was observed that the ABAI still achieved the highest classification accuracy in the uni-temporal classification results and had slightly lower accuracy than the dNBR in the bi-temporal classification results. This indicates that the ABAI also performs well in assessing fire severity using Landsat 8 OLI data. Apart from the influence of satellite sensors, fire scar areas are often subject to confusion with cloud cover, water bodies, terrain shadows, and other background noise, especially in complex environments [41–43]. Current research suggests that the ABAI demonstrates good resistance to interference when identifying fire scar areas in the presence of terrain shadows, cloud cover, water bodies, and other land cover types. Therefore, in fire-prone areas with complex terrain and shadow interference, the ABAI has a significant advantage in assessing fire severity.

Furthermore, based on the accuracy assessment confusion matrices for each experiment, it was found that, in most cases, the ABAI achieves the highest accuracy in classifying unburned and highly burned severity levels, but its accuracy is lower for low and moderate burn severity levels compared with other spectral indices. This is because, during the development of ABAI, a multi-objective optimization approach was used to enhance the discrimination between burned areas and other land cover types [44]. However, at that time, the classification of low and moderate burn severity levels was not considered, and this aspect will be improved in future research.

Certainly, our current work is preliminary, and there is much work to be done in the future. First, we have only validated the effectiveness of the ABAI in assessing fire severity using Sentinel-2 and Landsat 8 OLI satellite data; the application of other satellite sensors like MODIS, ETM+, and GF-1 has not been tested yet. Second, the ABAI has only conducted fire severity assessments in a relatively small area, discussing only cases in China and the United States. New issues may arise in other vegetation conditions, e.g., Australia and Africa. Its potential to assess fire severity in different climates and terrains needs further research. Additionally, since the ABAI is similar to other fire indices, it would be interesting to study its sensitivity in monitoring post-fire vegetation recovery trends, and this will be the subject of future research.

## 5. Conclusions

The results of the experiments conducted in different scenarios and study areas suggest the following: (1) The ABAI has a significant advantage in terms of accuracy and stability in assessing fire severity compared with other remote sensing indices like the NBR, the NDVI, and the MIRBI, especially in areas with significant topographic shadows. (2) When using uni-temporal remote sensing data, the ABAI index shows certain advantages in assessing forest fire severity, particularly in heavily burned areas, and it performed almost as well as the dNBR in bi-temporal remotely sensed data. (3) The ABAI performs superiorly with both Sentinel-2 and Landsat 8 OLI data, indicating that the ABAI we constructed is versatile and can be applied to different sensor image data.

Given the results of fire severity assessments across different scenarios and sensors, we conclude that the ABAI is a relatively effective remote sensing index for fire burn severity assessment with the potential to replace or complement existing remote sensing burn indices. However, despite the extensive validation this model has received, as a newly developed remote sensing index, its application effectiveness and scope require further rigorous and large-scale validation.

**Author Contributions:** Conceptualization, B.W.; methodology, B.W.; investigation, R.G. and B.W.; software, R.G.; resources, B.W.; writing—original draft preparation, R.G.; writing—review and editing, R.G., J.Y. and H.Z.; visualization, R.G.; supervision, B.W.; funding acquisition, B.W. All authors have read and agreed to the published version of the manuscript.

**Funding:** This work was funded by the Natural Science Foundation of China, grant number 42371419 and 41961055. This research was funded by the National Key R&D Program "Research on forest and grassland fire early warning and prevention technology and key equipment" (Project No. 2018YFE0207800).

**Institutional Review Board Statement:** Not applicable.

**Informed Consent Statement:** Not applicable.

**Data Availability Statement:** The Sentinel-2 remote sensing image datasets that support the findings of this study are available at https://sentinel.esa.int/web/sentinel/missions/sentinel-2, accessed on 9 March 2023. Landsat 8 OLI remote sensing images can be downloaded from the website https://glovis.usgs.gov/app, accessed on 9 March 2023.

**Acknowledgments:** The authors would like to thank the anonymous reviewers and editors for their valuable comments. We thank the ESA (European Space Agency) and the USGS (United States Geological Survey) for providing Sentinel-2 and Landsat 8 OLI data with free access.

**Conflicts of Interest:** The authors declare no conflicts of interest.

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
