# Peer review of "Assessment of the Analytic Burned Area Index for Forest Fire Severity Detection Using Sentinel and Landsat Data"

_fire, doi:10.3390/fire7010019_

Round 1
Reviewer 1 Report
Comments and Suggestions for Authors
The manuscript titled "Assessment of the Analytic Burned Area Index for Forest Fire Severity Detection using Sentinel and Landsat data" represents a validation of the Analytic Burned Area Index (ABAI) for assessing fire severity. Overall, it yields somewhat favorable results when compared to other algorithms such as NDVI and SAVI, which are not developed as burned indices, as well as the Normalized Burn Ratio (NBR) and Mid-Infrared Bi-spectral Index (MIRBI), designed to evaluate burned areas. It would be desirable to compare indices of the same type to guarantee the objectivity of the research.
The document, in general, is well-structured; however, some issues arise, particularly concerning the organization, especially the placement of figures. In certain instances, figures are either missing or not appropriately referenced in the document. The inclusion of this information would greatly enhance the comprehensibility of the research undertaken. For example, Figure 10, which is present in the document, requires further enhancement to facilitate a clearer understanding of the burn area patterns.
The methods section needs clarification. Firstly, Figure 2, indicating the flow chart of methods, is not present. Also, it is necessary to justify the use of spectral indices such as NDVI. In Tucker's research (1979), NDVI is not clearly defined as an index related to burn intensity. Please use up-to-date references.
In tables, it is recommended to arrange the information of the different indices in the same order. Furthermore, in Table 9, the comparison of the accuracy assessment of burn severity is performed using two satellite sensors with different characteristics. This comparison encompasses the environmental conditions of the two locations as well as differences in technology, given that Landsat differs from Sentinel 2.
Related to the results, the classifications' results are emphasized, selecting the highest accuracy, but the values and possible causes of the values are not discussed. Independently, for each sensor, the highest result is represented, but the index is still slow.
Reviewer 2 Report
Comments and Suggestions for Authors
Thank you for the invitation to review this manuscript. I have thoroughly examined the content presented in the paper and find myself reasonably content with the motivation and findings. The study explores a new fire severity index, named ABAI (published last year in the Forest journal by the same research team), utilizing Landsat-8 and Sentinel-2 images to establish its relationship with CBI. Additionally, the paper compares the performance of various indexes on both uni-temporal and bi-temporal images. The aim and contributions of the paper are clearly articulated, and I did not identify any significant issues in the empirical design. However, there are areas that require revision, and the authors need to clarify certain points before the paper can proceed to publication. Here are my observations:
1) Besides ABAI, the paper employs SAVI, NDVI, NBR, and MIRBI indexes for comparison. NDVI and SAVI are primarily vegetation health indexes, with their primary objective not being the detection of fire severity. They could be considered auxiliary indexes. However, some burn indexes, such as BAI, CSI, NBRSWIR, NBRT1-2-3, NDSWIR, SAVIT, VI6T, are not included in the regression analyses. Some of these indexes are becoming more common for delineating fire severity. The experimental design should have included a more extensive set of burn indexes, and the authors need to explain why they chose not to include a larger set of indexes for comparison.
2) The main issue in the paper is the disorganized flow of the results sections. The subsection on "Validation for Fire Severity Detection" involves S2 unitemporal images for China and L8 unitemporal images for the USA, using two different sensors for two different fire zones. The subsection on "The sensitivity of ABAI for fire severity" presents S2 bitemporal images for China and L8 bitemporal images for the USA. The subsection on "assessing the impact of different sensors" involves L8 unitemporal images and bitemporal images for China simultaneously. This demonstration is confusing and appears incompatible. If the authors intend to compare sensors, they should do so in the same study area. Similarly, if they want to compare the performance of an index in different geographic locations, they should keep the sensor conditions constant.
3) The regression plots indicate that a clear linear-polynomial fit cannot be guaranteed. The relationship between dependent and independent variables could have been investigated with non-linear regressors (e.g., random forests, support vectors, etc.). The authors need to elucidate the rationale behind selecting linear and lower-degree polynomial regressors, perhaps by including a correlation matrix or collinearity test between the variables.
4) The CBI data samples for Okanagon, USA, lack date information.
5) All S2 and L8 scenes should be listed with their IDs, acquisition dates, path/row, cloud cover information in a table in the materials section.
6) It would be preferable to present CBI samples on maps corresponding to the study areas. This would spatially represent the input data and is considered a requirement.
7) Did the authors resample L8 pixels to 10 meters, similar to what was done for S2?
8) Have the authors considered making the data available as open access to enhance impact and facilitate reusability?
9) The authors should address the management of slightly different wavelengths between L8 and S2. For example, the SWIR2 band of L8 covers a wavelength window of 2107-2294 nm, whereas S2 has a wavelength of 2202.4 nm. Similar variations exist between the SWIR1 and Green bands.
Reviewer 3 Report
Comments and Suggestions for Authors
Rentao Guo submitted the article Assessment of the Analytic Burned Area Index for Forest Fire Severity Detection using Sentinel and Landsat data to mdpi fire for review. The author evaluated a newly published Analytic Burned Area Index (ABPI) originally developed to map burned areas here tested for Severity Detection. The author tested 10 indexes including ABPI to test burning index detection from remote sensing data. Author provides a clear explanation of the method and objectives of the study.
Cases used to validate the model remain limited as indicated by the authors. Nevertheless the paper is relatively well constructed with logical development and conclusion supported by the results obtained as the study consists of a preliminary limited study of the potential of ABAI.
Questions
-1 Is there a better/alternative way to construct an index that combined both R2 and RMSE pro/con.
-2 Looking at the spread of data Fig 3 for example and the attempt to have data fitting. Is there any option to have a multi index super index to improve accuracy ?
-3 Should not the score combined all the cases in a summary results. Why Ganzhou and Okanogan values are comuted separately ? How do you compare Ganzhu that contains 3 study areas and Okanogan that has only one ?
-4 Do you expect impact of the latitude on the quality of index or impact of the season
-5 what caused a detection of high severity for the dMIRBI model Figure 8 as it largely overestimated the burned area.
Detail comments
Explain the Kappa coefficient before using Line 264. Have you explained its utility ?
Summary and Discussion
Precise the number of study sites used to validate the models, 240 CBI January 2021 data in Ghanzou separated in 3 study areas and 257 CBI in July 2014. You may precise for each the number of validated points used meaning the gridded observation.
You may indicate that the analysis only focuses on China and US cases. There may be new issues in other vegetation situations, for example Australia, Africa.
Figures
Figure 1
Similarely to (b), drop one of the Jiangxi Province map to add one of China with the location of Jiangxi Province on it
Tables
Table4
You may underline or highlight the best score obtained per categories and Study Area for fast understanding.
Round 2
Reviewer 2 Report
Comments and Suggestions for Authors
Thank you for inviting me to assess the revised manuscript. Having thoroughly examined the revised document and its accompanying response letter, I observed that the authors endeavored to incorporate additional indices for comparison, restructured the results section to alleviate complexity, and addressed various minor issues. I find no additional comments to make at this time.
Reviewer 3 Report
Comments and Suggestions for Authors
The authors replyed to all the questions systematically.
Th quality of the paper have improved.